# Transcriptional Insights of Oxidative Stress and Extracellular Traps in Lung Tissues of Fatal COVID-19 Cases

**DOI:** 10.3390/ijms24032646

**Published:** 2023-01-31

**Authors:** Aref Hosseini, Darko Stojkov, Timothée Fettrelet, Rostyslav Bilyy, Shida Yousefi, Hans-Uwe Simon

**Affiliations:** 1Institute of Pharmacology, University of Bern, 3010 Bern, Switzerland; 2Department of Histology, Cytology and Embryology, Danylo Halytsky Lviv National Medical University, Pekarska Str. 69, 79010 Lviv, Ukraine; 3Institute of Biochemistry, Brandenburg Medical School, 16816 Neuruppin, Germany

**Keywords:** COVID-19, RNA-seq, neutrophil extracellular traps, ROS, oxidized DNA, lipid peroxidation

## Abstract

Neutrophil extracellular traps (NETs) and oxidative stress are considered to be beneficial in the innate immune defense against pathogens. However, defective clearance of NETs in the lung of acute respiratory syndrome coronavirus 2 (SARS-CoV-2)-infected patients could lead to severe respiratory syndrome infection, the so-called coronavirus disease 2019 (COVID-19). To elucidate the pathways that are related to NETs within the pathophysiology of COVID-19, we utilized RNA sequencing (RNA-seq) as well as immunofluorescence and immunohistochemistry methods. RNA-seq analysis provided evidence for increased oxidative stress and the activation of viral-related signaling pathways in post-mortem lungs of COVID-19 patients compared to control donors. Moreover, an excess of neutrophil infiltration and NET formation were detected in the patients’ lungs, where the extracellular DNA was oxidized and co-localized with neutrophil granule protein myeloperoxidase (MPO). Interestingly, staining of the lipid peroxidation marker 4-hydroxynonenal (4-HNE) depicted high colocalization with NETs and was correlated with the neutrophil infiltration of the lung tissues, suggesting that it could serve as a suitable marker for the identification of NETs and the severity of the disease. Moreover, local inhalation therapy to reduce the excess lipid oxidation and NETs in the lungs of severely infected patients might be useful to ameliorate their clinical conditions.

## 1. Introduction

Coronavirus disease 2019 (COVID-19), caused by the severe acute respiratory syndrome coronavirus 2 (SARS-CoV-2), has resulted in a global pandemic with significant morbidity and mortality worldwide, with the elderly population at particular risk for severe disease and mortality [1]. Neutrophils play a crucial role in viral clearance in terms of neutrophil extracellular traps (NETs) and the production of type 1 interferons (IFN-α/IFNβ) [2]. On the other hand, exacerbation of neutrophils’ effector function could also have detrimental effects on the pathogenesis of SARS-CoV-2, aggravating the complications of COVID-19 [3].

NET formation is an important effector function of neutrophils to immobilize and kill invading microorganisms in the extracellular environment [4]. Emerging evidence suggests that excessive NET formation and delay in NETs’ degradation contribute to inflammation, organ damage, immunothrombosis, and vascular occlusion that characterize the clinical conditions of severe COVID-19 patients [5,6,7,8]. It should be noted that inflammatory immune responses in the lungs of chronic obstructive pulmonary disease (COPD) [9], bronchial asthma [10], and cystic fibrosis patients [11,12] have also been associated with NET formation. In severe cases of COVID-19, pneumonia progresses to respiratory failure, and excess NET formation further propagates inflammation and microvascular thrombosis in the lungs, leading to acute respiratory distress syndrome (ARDS) [13]. Moreover, plasma markers associated with NET formation have been correlated with disease severity in COVID-19 patients, implicating a role for neutrophils in the pathology of this disease [14]. In addition, NET-containing microthrombi have been detected in individuals who died owing to SARS-CoV-2 infection [15]. Additional reports have been published that suggest that both ARDS and acute lung injury are associated with excessive NET formation in SARS-CoV-2 infections [16,17,18]. NET formation also contributes to SARS-CoV-2-induced cytokine storm (CS) [19] and is associated with immuno-thrombosis [20]. There seems to be a delicate balance between NET formation and the progression of pulmonary function in COVID-19 patients’ lungs [21]. NETs have also been associated with a higher risk of morbid thrombotic events despite prophylactic anticoagulant treatment [22]. Therefore, a potential therapy to block the excess NETs and subsequently alleviate SARS-CoV-2 infection-exacerbated immune response has been suggested [20,23,24].

There is little information available regarding the mechanism that leads to immune cell activation, tissue damage, and thrombosis in patients suffering from severe COVID-19 disease. However, the proximity of the released NET components (e.g., NE, MPO, and dsDNA) that mediate a direct cytotoxic impact on the alveolar epithelium and endothelium, resulting in loss of alveolar integrity, has been suggested [25]. Moreover, NETs might suppress the adaptive arm of the immune system, i.e., T cells that are necessary to kill virus-infected cells [26]. These reports support the notion that continuous NET formation is involved in disease progression and point to the importance of the direct investigation of lung tissues in order to obtain insights regarding the molecular pathways driving disease exacerbation. It should be noted that most studies have analyzed temporal changes in the plasma of COVID-19 patients to detect NETs. In contrast, there is only little information available from lung studies.

Inflammation is one of the most essential and beneficial defense mechanisms of the host, but it is also one of the most common ways whereby tissues are injured [27]. Reactive oxygen species (ROS) are key signaling molecules that play an essential role in the progression of inflammation [28]. ROS could target lung epithelial cells as a primary target and can cause oxidative deoxyribonucleic acid (DNA) modification, such as 8-hydroxy-deoxyguanosine (8-OHdG) [29]. In both nuclear and mitochondrial DNA, the 8-OHdG modification is one of the predominant forms of free radical-induced oxidative lesions and has accordingly been widely used as a biomarker for oxidative stress [30]. Moreover, mitochondrial ROS (mtROS) can also promote tissue injury by repressing mitochondrial transcription factor A (TFAM)-mediated mitochondrial DNA (mtDNA) maintenance, resulting in decreased mitochondrial energy metabolism and increased cytokine release [31]. Neutrophils can release oxidized mtDNA [32,33], and damaged mitochondria release intact or oxidized mtDNA, which promotes spontaneous innate immune responses [34].

Previous research has suggested that the most pathological host responses to SARS-CoV-2 infection that leads to COVID-19 lung damage are due to the hydroxyl radical (•OH) that associates with ferric cation (Fe^3+^) within the hemoglobin heme group. A higher oxidation state of the iron molecule makes it unable to bind with oxygen, resulting in less efficient oxygen transport despite a high oxygen supply [35]. Accordingly, radical-scavenging agents might be suitable therapeutic choices [36,37,38,39,40]. Interestingly, a direct correlation between COVID-19 and viral-induced ROS was detected in the sputum of COVID-19 patients using an electrochemical sensor [41].

Lipid peroxidation (LPO) is one of the most critical events mediated by ROS, contributing to the pathophysiology of human health and disease [42]. LPOs have a longer lifetime than ROS and represent better biomarkers of SARS-CoV-2-induced oxidative stress. The final products of LPOs are reactive aldehydes, and 4-hydroxynonenal (4-HNE) is one of the most bioactive LPO-derived reactive aldehydes [43]. A previously published work has suggested high levels of the 4-HNE in the plasma of deceased COVID-19 patients [44]. Moreover, LPOs depicting oxidative stress markers have been demonstrated on formaldehyde-fixed, paraffin-embedded tissue sections from post-mortem tissue samples by immunohistochemistry [43,44]. 

To study the role of NETs in COVID-19 disease, most of the studies quantified the levels of cell-free DNA, myeloperoxidase-DNA (MPO-DNA), and citrullinated histone H3 (Cit-H3) in the sera of patients. Of note, dead cells could also release DNA and Cit-H3, especially when phagocytosis is impaired [13]. Therefore, the presence of these markers in blood or sputum is not an exact indication of NET formation in COVID-19 patients. So far, very few reports have combined transcriptional analysis using RNA-seq and immuno-fluorescence/immunohistochemistry staining to analyze direct ROS/NETs activity under in vivo conditions. Here, we demonstrate an enriched NET formation pathway combined with LPO-derived reactive aldehydes and oxidized DNA damage in the lungs of severe COVID-19 patients. It is suggested that the excess of neutrophil infiltration, NET formation, and reactive oxygen species results in tissue destruction and organ dysfunction in fatal cases of COVID-19 patients.

## 2. Results

### 2.1. Lung Transcriptomes Reveal Upregulation of Inflammatory Immune Response Genes in Fatal Cases of COVID-19 Patients

To better understand the molecular basis underlying fatal COVID-19 cases, we performed total RNA-seq on the post-mortem formalin-fixed paraffin-embedded (FFPE) lung samples from seven SARS-CoV-2-infected patients. As controls, we used histologically normal lung tissues obtained from three uninfected donors who were deceased due to sudden death unrelated to viral infection or respiratory diseases. Principal component analysis (PCA) was applied to show the separation of COVID-19 and control samples by two-dimensional data representation (Appendix A). Differential expression analysis by DESeq2 identified 272 up-regulated and 202 down-regulated genes that were compared among COVID-19 patients and control donors (adjusted *p* value < 0.05; log2FC > |2|) (Figure 1a and Appendix A) (GEO accession GSE208076). Of note, FcR for IgA, *FCRLA*, was one of the most up-regulated genes (log2FC = 5.9) that induced NETs more potently than Fc gamma receptors [45]. Heightened expression was further found for the pathological extracellular matrix (*COL1A1*), which was reported to be up-regulated in response to increased ROS and DNA damage [46]. Further, antigen recognition by B cells, immunoglobulin-heavy variable (*IGHV*)*1-24*, *IGHV3-20*, *IGHV4-34*, *IGHV2-26*, and *IGLV3-19*, and the scavenger receptor cysteine-rich family member with 5 Domains (*SSC5D*) were up-regulated. Down-regulated genes showed lower fold changes than the up-regulated genes, including cartilage acidic protein 1, *CRTAC1*, (log2FC = −4.16), previously known to be down-regulated in COVID-19 patients with severe infection [47]. Differentially expressed gene (DEG) expression levels in each sample are visualized in a heatmap, and samples are clustered based on Euclidean distance as shown in Figure 1b. Finally, to confirm the RNA-seq data, qPCR analysis was performed to assess the relative mRNA expression of *COL1A1* as highly up-regulated and *CRTAC1* as down-regulated genes in patients. Indeed, the mRNA expression analyses by qPCR confirmed the RNA-seq data (Figure 1c). Collectively, the RNA expression data suggest a strong immune response and increased oxidative DNA damage and signaling molecules involved in NET formation associated with COVID-19.

### 2.2. Molecular Signaling Pathway Enrichment Analysis Proposes Possible Inflammation and Neutrophil Extracellular Trap (NET) Formation in the Lungs of COVID-19 Patients

Gene Set Enrichment Analysis (GSEA) was performed on the normalized expression value of genes. GSEA was conducted with the hallmark gene set collection. Enriched signaling pathways with normalized *p* value < 0.001 were considered, including oxidative phosphorylation and ROS pathways (Figure 2a and Appendix A). In addition, functional analysis of the up-regulated genes by Enrichr, another enrichment analysis tool, showed that NET formation is among the top 10 enriched pathways in COVID-19 patients (Figure 2b). The list of up-regulated genes involved in NET formation is depicted as a heatmap in Figure 2c. 

### 2.3. Neutrophil Extracellular Traps (NETs) in the Lungs of COVID-19 Patients Are Associated with Oxidized dsDNA

The lung tissues were stained to detect the presence of extracellular DNA and neutrophil granule proteins, such as myeloperoxidase (MPO). As expected, SARS-CoV-2-infected patients’ lung tissue sections revealed high neutrophil infiltration with strong NET formation, but this was not present in control donors (Figure 3a,b). Quantification of the total infiltrating neutrophils in the lung tissues performed by an automatic digital slide scanner revealed evidence for increased neutrophil infiltration in the lungs of COVID-19 patients compared to control donors (Figure 3a,c). Activated neutrophils release oxidized mtDNA in a ROS-dependent manner [32,33]. To investigate the presence of oxidized DNA, we used confocal microscopy and analyzed the lung tissues from COVID-19 patients and control donors. The extracellular structures consisted of neutrophil granule proteins (MPO) co-localized with oxidized DNA (8-OHdG), which was absent in control lung tissues (Figure 3b). The staining intensity of the oxidized DNA marker (8-OHdG) was significantly higher in COVID-19 patients compared to control lung tissue and showed a positive correlation with the MPO (Figure 3d). These results demonstrate the presence of NETs that exhibit an oxidized DNA scaffold in the lungs of COVID-19 patients. 

### 2.4. Lipid Oxidation Occurs in the Lungs of COVID-19 Patients, and 4-Hydroxynonenal (4-HNE) Co-Localizes with NETs

Direct measurement of ROS levels with high accuracy and precision is difficult in tissues owing to their short lifespan and rapid reactivity with redox state-regulating components [48]. In addition, no report has directly demonstrated the presence of ROS activity in COVID-19 lung tissues, perhaps due to the biochemical instability of ROS molecules upon formalin fixation [49]. On the other hand, lipid peroxidation-derived aldehydes, such as 4-hydroxynonenal (4-HNE), are shown to be more stable in formalin-fixed tissues that are exposed to high ROS activity [50]. Lung tissues of COVID-19 patients and control donors were investigated for the presence of lipid peroxidation using immunohistochemistry as previously described [51]. In this study, the paraformaldehyde-fixed lung tissue sections were subjected to 4-HNE immunochemistry and counterstained with hematoxylin/eosin (H&E) staining. The COVID-19 lung tissues depicted extensive lipid oxidation compared to control donors (Figure 4a). To investigate the correlation of lipid oxidation with neutrophil infiltration and NET formation in the lungs of COVID-19 patients, an immunofluorescence staining method was employed to detect the co-localization of NETs (MPO and DNA) together with the LPO marker 4-HNE using confocal microscopy (Figure 4b). NETs contained extracellular dsDNA decorated with MPO, which strongly co-localized with 4-HNE (Figure 4b). Evidence for the presence of 4-HNE in NETs is also shown in Appendix A, depicted as a z-stack. The total number of neutrophils infiltrating the lung tissues was quantified by an automatic digital slide scanner. The results confirmed that neutrophil numbers in COVID-19 lung tissues are increased (Figure 4c). Quantification of LPOs’ staining intensity by Imaris software revealed significant differences between the COVID-19 patient and control donor lung tissues (Figure 4d). Furthermore, a high positive correlation between lipid peroxidation and MPO intensity was observed in lung tissues of COVID-19 patients (Figure 4d). These results indicate that 4-HNE could be a suitable marker to depict NETs and suggest that the presence of lipid peroxidation combined with an excess of NETs might be suggestive for the presence of tissue damage.

## 3. Discussion

We have performed a detailed mRNA and protein expression analysis using RNA-seq as well as immunofluorescence and immunohistochemistry analyses of post-mortem biopsies of lung tissues from COVID-19 patients and control donors. The clinical and immunological features of patients with COVID-19 have highlighted a potential role of an immune cell-mediated pathology of the lungs of COVID-19 patients. Here, we demonstrate that the pathways related to NET formation and oxidative stress are among the major up-regulated transcriptional signatures in lung tissue obtained from fatal cases of COVID-19 patients. Further, 8-OHdG is a specific indicator of DNA damage caused by ROS activity [33], and 4-HNE is an effective bioactive indicator of lipid peroxidation caused by oxidative stress [52]. Previously published works have suggested that 8-OHdG is a prognostic biomarker for COVID-19 that can be quantified in urine and blood [53,54]. However, in this study, we demonstrate for the first time the direct evidence for the association of MPO, a highly versatile oxidative enzyme, with a high level of 8-OHdG-labeled extracellular DNA. Moreover, we show evidence for an excess of 4-HNE that co-localized with NETs in lung tissues of post-mortem COVID-19 patients (Figure 3 and Figure 4).

ROS induction as a potential therapeutic measure against viral infections has been proposed, and it was hypothesized that ROS-based treatment could impair RNA integrity faster than other macromolecules [55]. Inversely, it has been reported that the mortality of patients may be attributed to the excess of ROS activity owing to their immune response [56]. Moreover, higher lipid peroxidation levels were associated with a higher risk of hospitalization or death in COVID-19 patients [57]. In addition, a distinct metabolic signature and enhanced neutrophil activation, together with high ROS generation, were associated with severe symptoms in COVID-19 and a transient interferon-stimulated gene signature [58]. Immune responses associated with “cytokine storms” [59] could activate ROS, which may result in the consumption of nitric oxide (NO), a critical vasodilation regulator. Neutrophils execute their microbicidal activity mainly through reactive halogen species (RHS) and ROS generation catalyzed by MPO [60]. We used the term “ROS” throughout the manuscript instead of ROS and RHS for simplicity. Activated neutrophils are known to release MPO in a natural immune response, which contributes to the production of hypochlorous acid (HOCl) [61], protein nitration, protein oxidation, lipid peroxidation, S-nitrosylation, and DNA damage [62]. Additionally, when extracellular ROS generation and pro-inflammatory cytokine generation are significantly up-regulated, tissue damage and perpetuating inflammation are likely to occur under such conditions [63]. On the other hand, ROS are known to function as signaling molecules to regulate certain cellular processes under physiological conditions. Therefore, utilizing general antioxidants may interfere with the advantageous effects of ROS that are necessary for optimal physiological cellular functions [64]. Consequently, it was recommended that inhibitors focusing on oxidant generation from particular sources should be used to overcome issues with antioxidant therapy [65].

Although ROS are normally produced for specific cellular functions, once liberated in the cytoplasm, they can interact with proteins and lipids, leading to damaged cell structures. Over time, this oxidative damage to the cellular machinery compromises the cell’s ability to operate properly. Additionally, ROS rapidly attack DNA, resulting in lesions in the genetic code with the likelihood of mutations [66].

The evaluation of different transcriptome data sets shows that SARS-COV-2 induces the expression of oxidative stress genes via both immune and lung structural cells. Therefore, targeting oxidative stress genes has been proposed as a practical therapeutic approach for treating COVID-19 patients [67]. Our RNA-seq data analysis revealed the enrichment of signaling pathways implicated in oxidative phosphorylation, which is required for NET formation and ROS production in the lungs of fatal cases of COVID-19 patients. Previously, it has been reported that 4-HNE co-localizes with MPO in infarcted brains of mice, suggesting that oxidants produced by MPO form the oxidative products [68]. *N*-acetyl lysyltyrosylcysteine amide (KYC), an MPO-specific inhibitor, has been shown to effectively reduce oxidative stress-mediated brain injury after stroke by reducing 4-HNE and other toxic oxidants in vivo [68]. 

In this study, we also observed the co-localization of MPO and 4-HNE, as well as 8-OHdG in the lung tissues of deceased COVID-19 patients, indicating that MPO-mediated oxidative stress may also play a crucial role in the lung injury of COVID-19 patients with severe infection. Accordingly, KYC could be considered for therapeutic lung inhalation in these patients. Additionally, we report that in the lungs of post-mortem COVID-19 patients, MPO positively correlates with oxidized DNA and lipid peroxidation. We observed a stronger correlation with 4-HNE than with 8-OHdG. Therefore, among these two molecules, 4-HNE might be a better marker for the detection and quantification of NETs in plasma.

Our study has some limitations. For instance, we utilized FFPE samples and not fresh tissues from COVID-19 patients. The uncertainty regarding the fidelity of FFPE RNA is a limitation. Furthermore, only fatal cases of COVID-19 patients and control donors have been examined in this study. Less severely affected COVID-19 patients were not investigated. In addition, the size of the cohort was rather small, and different time points during disease progression have not been studied.

In summary, our findings deepen our understanding of oxidative stress in SARS-CoV-2 infection and support its critical role in COVID-19. The results indicate the enrichment of signaling pathways involved in oxidative phosphorylation required for the generation of ROS and NET formation in patients under in vivo conditions. Moreover, NETs were co-localized with oxidized DNA and lipid oxidation, and the quantification of lipid oxidation demonstrated a high correlation with the amount of neutrophil infiltration. Therefore, our data provide a link between high levels of MPO and increased lipid oxidation at the site of inflammation, pointing to the possibility that LPOs might be suitable new drug targets in future therapeutic approaches for COVID-19 patients with severe infection.

## 4. Materials and Methods

### 4.1. RNA Isolation and RNA Sequencing

Total RNA was isolated from formalin-fixed, paraffin-embedded (FFPE) lung tissues of seven patients and three non-covid-infected individuals using the Maxwell^®^ RSC RNA FFPE Kit in the Maxwell^®^ RSC Instrument (Promega, Dübendorf, Switzerland). The quantity and quality of the extracted RNA were assessed using a ThermoFisher Scientific Qubit 4.0 fluorometer with the Qubit RNA BR Assay Kit (ThermoFisher Scientific, Q10211, Basel, Switzerland) and an Advanced Analytical Fragment Analyzer System using a Fragment Analyzer RNA Kit (Agilent Technologies AG, DNF-471, Basel, Switzerland), respectively. The DV200 was determined for each RNA sample to assess the quality of these FFPE samples better and to adjust the library preparation protocol with regard to fragmentation time and the clean-up steps [69]. 

Ribosomal RNA was depleted from 100 ng of total RNA samples using a RiboCop for Human/Mouse/Rat (HMR) V2 Kit (Lexogen, 144, Vienna, Austria) according to the vendor guidelines. Thereafter, sequencing libraries were produced using a Swift RNA Library Kit (Swift Biosciences, 10009985, Ann Arbor, MI, USA) in combination with Swift Unique Dual Indexing Primers (Swift Biosciences, 10009912) and according to a technical note developed by Swift Biosciences entitled “The Swift RNA Library Kit Optimizes RNA-Seq Data Quality and Costs for FFPE RNA”.

Pooled complementary DNA (cDNA) libraries were paired-end sequenced using one lane of an Illumina NovaSeq 6000 S1 Reagent Kit v1.5 (200 cycles; illumina, 20028318, San Diego, CA, USA) on an Illumina NovaSeq 6000 instrument, generating an average of 91 million reads/sample. The quality of the sequencing run was assessed using Illumina Sequencing Analysis Viewer (illumina version 2.4.7), and all base call files were demultiplexed and converted into FASTQ files using Illumina bcl2fastq conversion software v2.20. The quality control assessments, library generation, and sequencing run were performed at the Next Generation Sequencing Platform, University of Bern.

### 4.2. RNA-Sequencing Analysis

The quality of the RNA-seq data was assessed using Fastqc v. 0.11.9. The reads were mapped to the reference genome using HISAT2 v. 2.2.1 [70]. FeatureCounts v. 2.0.1 [71] was used to count the number of reads overlapping with each gene, as specified in the genome annotation (Homo_sapiens.GRCh38.94). Subsequent data analysis was performed in R v. 4.1.0. The Bioconductor package DESeq2 v. 1.32.0 [72] was used to analyze differentially expressed genes with default parameters. EnhancedVolcano v. 1.10.0 [73], pheatmap v. 1.0.12 [74], and ggplot2 v. 3.3.3 [75] were utilized to visualize the results. Further bioinformatics analyses were performed with gene set enrichment analysis (GSEA) [76] and Enrichr [77].

### 4.3. Complementary DNA Synthesis and Quantitative PCR

Quantitative PCR (qPCR) was performed to validate RNA-seq data. For this, cDNA was synthesized using 100 ng RNA with iScript™ gDNA Clear cDNA Synthesis Kit (Bio-Rad Laboratories AG, Fribourg, Switzerland). The transcription levels were measured in duplicate using iTaq™ Universal SYBR^®^ Green Supermix and specific primer pairs (Table 1) in the T100 Gradient Thermal Cycler (Bio-Rad Laboratories AG). 

#### 4.3.1. Immunohistochemistry

Paraffin-embedded tissue sections were deparaffinized and rehydrated with graded ethanol dilutions. Antigen retrieval was applied to the tissue sections in a pressure cooker using an Antigen Retrieval Machine (2100 Retriever) with EDTA Buffer pH 9. Immunohistochemistry was performed using the Dako REAL Detection Kit System, Alkaline Phosphatase/RED, Rabbit/Mouse Kit (Dako Cat.# S2022, distributed by Agilent Technologies AG) according to the manufacturer’s instructions and as previously described [78]. For washing steps, the Dako washing buffer diluted with 1:10 with distilled water was used, and the Dako dual endogenous enzyme block (S2003) was applied to the tissue section before staining to inhibit endogenous phosphatase activity. Tissue sections were incubated with mouse monoclonal anti-human 4-HNE (1:50 dilution; clone 12F7, Invitrogen cat. #MA5-27570, distributed by ThermoFisher Scientific) in Dako antibody diluent S2022, followed by a biotin-streptavidin system coupled with alkaline phosphatase and counterstained with Hematoxylin dye (Dako Cat. #S2020) and mounted with Aquatex (Merck Cat. #1.08562.005, distributed by Sigma-Aldrich, Buchs, Switzerland). The images were visualized using a light microscope (Carl Zeiss Micro Imaging, Jena, Germany). 

#### 4.3.2. Immunofluorescence and Confocal Microscopy

Paraffin-embedded tissue sections were deparaffinized and rehydrated with graded ethanol dilutions. Antigen retrieval was applied to the tissue sections in a pressure cooker using Tris-EDTA buffer (10 mM Tris-HCl and 2 mM EDTA, pH 9.0, plus 0.01% saponin). The nonspecific binding was blocked by preincubation of the tissue samples with a blocking buffer containing pooled human immunoglobulins, normal goat serum, and 7.5% BSA in PBS) at room temperature for 1 h. Indirect immunofluorescence staining was performed by incubating the paraffin sections overnight at 4 °C with mouse monoclonal anti-oxidized DNA/RNA damage antibody (1:500 dilution; clone 15A3, StressMarq Biosciences cat# AB5830, Victoria, BC, Canada); and polyclonal rabbit anti-human MPO (1:3000 dilution; DAKO cat# A0398, distributed by Agilent Technologies AG); and/or monoclonal mouse anti-4-hydroxy-2-nonenal (4-HNE) (1:150 dilution; clone 12F7, ThermoFisher Scientific cat #MA5-27570). Probes were washed with PBS, and secondary antibodies Alexa Fluor^®^ 488—conjugated goat anti-rabbit (1:400 dilution) or Alexa Fluor^®^ 545—conjugated goat anti-rabbit (1:400 dilution) and Alexa Fluor^®^ 488—conjugated goat anti-mouse (1:400 dilution) or Alexa Fluor^®^ 488—conjugated goat anti-mouse antibodies (ThermoFisher Scientific) were applied, and tissue samples incubated at room temperature for 1 h. Nuclei were further stained with Hoechst 33342 (5 mM). In some experiments, to depict the NETs, DNA was stained with propidium iodide (1 μg/mL) solution for an additional 10 min. Samples were washed and mounted in Prolong Gold mounting medium, and image acquisition was performed using confocal laser scanning microscopy LSM 810 (Carl Zeiss Micro Imaging) with a 63× or 40×/1.40 oil DIC objective and analyzed with IMARIS software, version 9.9.1 (Oxford Instruments, Zurich, Switzerland). For better visualization, the gamma correction function, together with min/max thresholds of Imaris software, were used to optimize the image display by intensifying the grey values. For overall qualitative analysis, stained tissues were scanned using an automated slide scanner (3DHISTECH slide scanner, Quant Center software, using Cell Count module, Budapest, Hungary), and infiltrating neutrophils were counted in 10 high-power fields (hpf) of highest activity. 

#### 4.3.3. Statistical Analyses

GraphPad Prism software version 8.0.1 (GraphPad Software Inc., La Jolla, CA, USA) was used to assess the data. A two-tailed Student’s *t*-test was utilized to compare groups, and the Pearson correlation coefficient was applied as a statistic for quantifying co-localization. All values are represented as mean ± SEM, and *p* < 0.05 was considered statistically significant. 

## Figures and Tables

**Figure 1 ijms-24-02646-f001:**
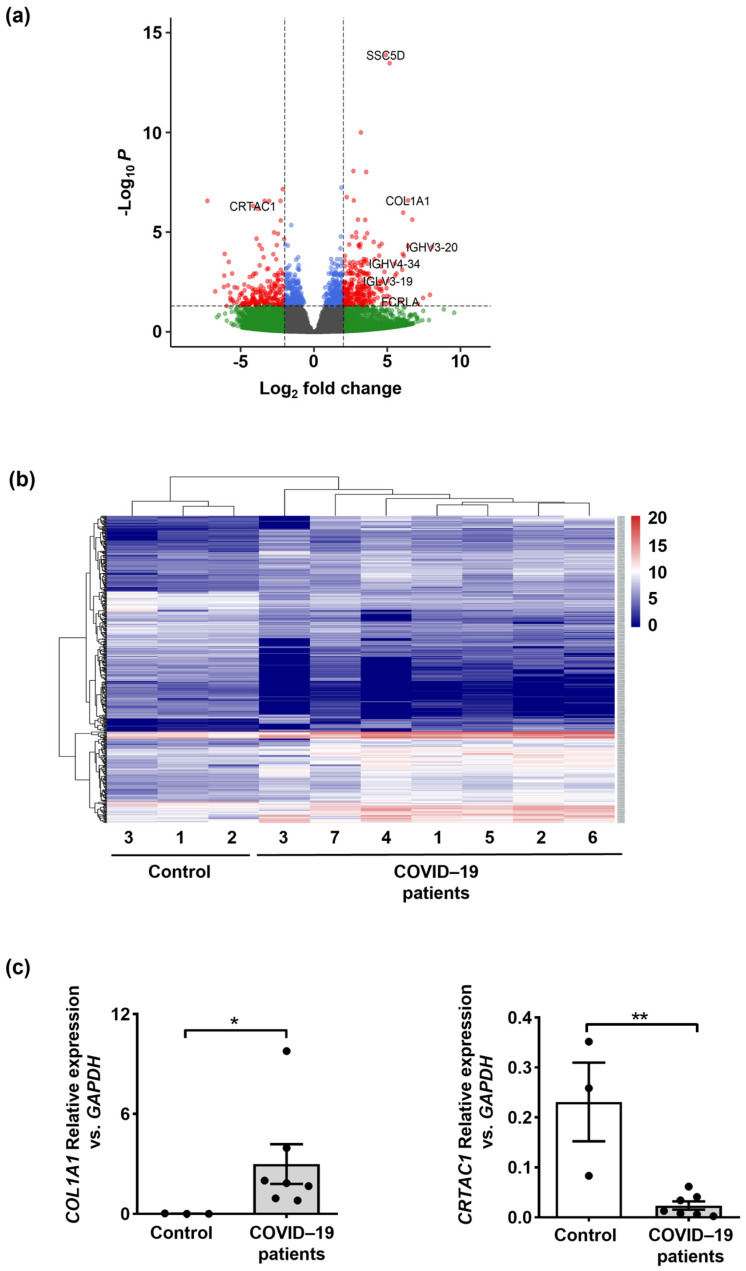
Expression analysis of the lung biopsies from COVID–19 patients and control individuals. (**a**) The volcano plot of differentially expressed genes (DEGs) in COVID–19 patients’ lungs was compared to control donors. A total of 474 DEGs were identified. (**b**) The heatmap of DEGs identified on lung tissue of COVID–19 patients (*n* = 7) compared to control donors (*n* = 3) for the expression pattern of the selected genes. (**c**) *COL1A1* and *CRTAC1* mRNA expression levels in lung tissues of COVID–19 patients (*n* = 7) compared to control donors (*n* = 3) by qPCR confirmed the RNA-seq data. Values are means ± SEM. * *p* < 0.05; ** *p* < 0.01.

**Figure 2 ijms-24-02646-f002:**
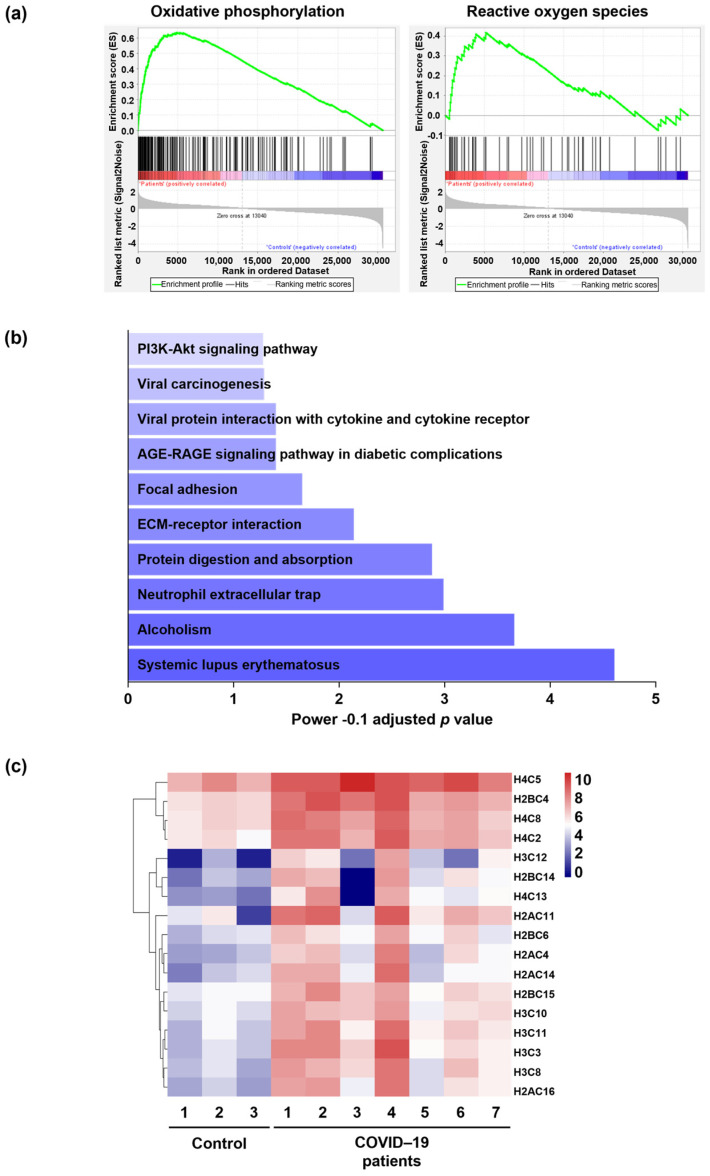
Analysis of enriched pathways. (**a**) Oxidative phosphorylation and ROS pathways were among enriched signaling pathways with a normalized *p* value < 0.001 in COVID–19 patients versus control lung tissues. Enrichment analysis was performed with the hallmark gene set collection based on ranked lists of all available genes by GSEA. The most enriched pathway is shown at the bottom of the graph, with a higher normalized enrichment score. (**b**) Top 10 enriched pathways in COVID–19 patients compared to control donors’ lung tissues based on Enrichr database. (**c**) Up-regulated genes (adjusted *p* value < 0.05 and log 2–fold change > 2) involved in NET formation depicted by heatmap in COVID–19 patients (*n* = 7) compared to control lung tissues (*n* = 3). The red color denotes higher, and the blue color indicates lower gene expression level.

**Figure 3 ijms-24-02646-f003:**
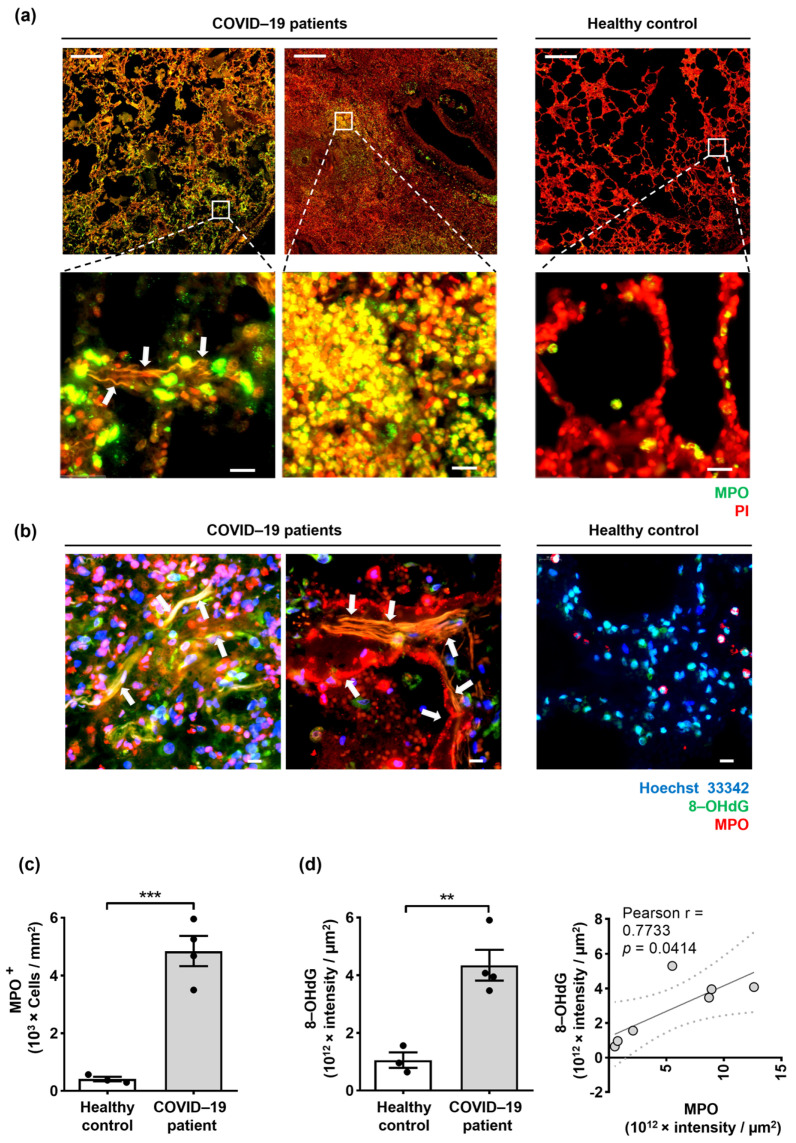
Neutrophil extracellular traps (NETs) in the lungs of COVID–19 patients are associated with oxidized dsDNA. (**a**) Lung tissues of COVID–19 patients (*n* = 4) and control donors (*n* = 3) were stained with anti-MPO antibody (green) and PI (propidium iodide, red) to detect NETs. Scale bars, 500 μm. Representative original data are shown. Lower panels: higher magnifications are shown with white arrows indicating the co-localization of extracellular DNA and MPO. Scale bars, 20 μm. Representative original data are shown. (**b**) Confocal microscopy of NETs is depicted with anti-MPO antibody (red), anti-8–OHdG antibody (green), and counterstained with Hoechst 33342 (blue) (colocalization is shown with white arrows). Scale bars, 10 μm. Representative original data are shown. (**c**) Neutrophil quantification in lung tissue of COVID–19 patients (*n* = 4) and control donors (*n* = 3) using an automatic digital slide scanner (Panoramic MIDI II, 3DHistech, Budapest, Hungary) is shown as cells/mm^2^. (**d**) Quantification of fluorescence intensity sums of MPO and 8–OHdG was performed for each tissue section using Imaris software, version 9.9.1 (Oxford Instruments, Zurich, Switzerland) as intensity sum/µm^2^. The right panel shows the correlation between the intensity of MPO and 8–OHdG in the lungs of COVID–19 patients (*n* = 4) and control donors (*n* = 3). Values are means ± SEM. Single data are presented in scatter dot plots. ** *p* < 0.01; *** *p* < 0.001.

**Figure 4 ijms-24-02646-f004:**
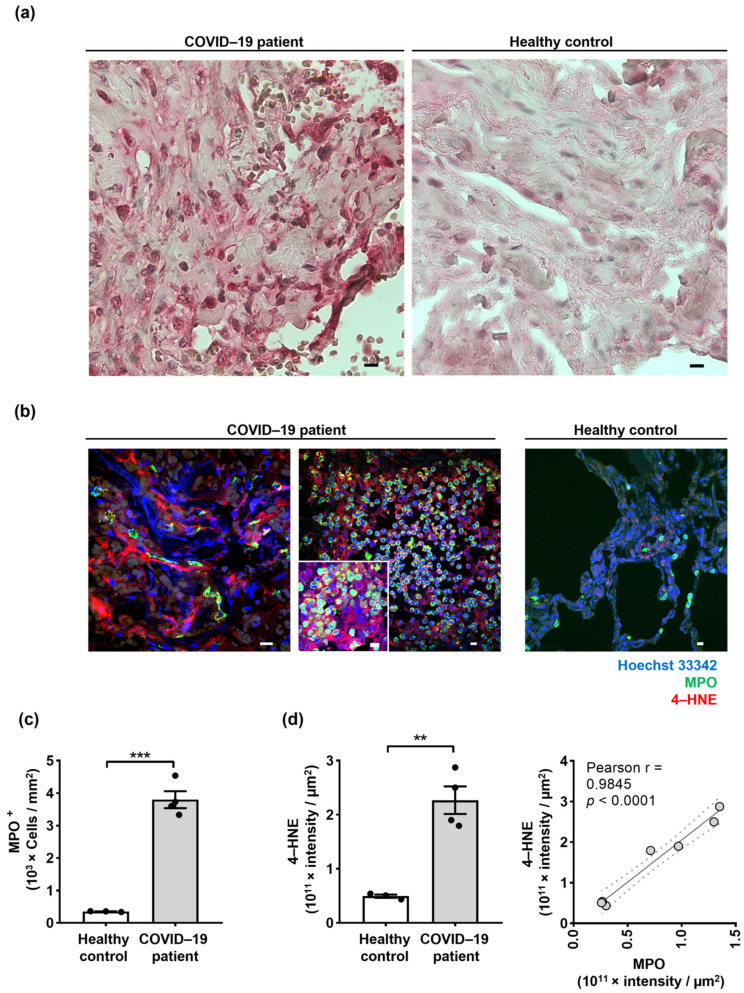
Increased lipid oxidation occurs in the lungs of COVID–19 patients and its possible involvement in NET formation. (**a**) Paraformaldehyde-fixed lung tissues of COVID–19 patients (*n* = 4) and control donors (*n* = 3) were subjected to 4–HNE antibodies by immunohistochemistry. Scale bars, 20 μm. Representative original data are shown. (**b**) Confocal microscopy analysis of NETs in COVID–19 patients (*n* = 4) and control donors (*n* = 3) depicted by anti-MPO antibody (green), anti-4–HNE antibody (red), and counterstained with Hoechst 33342 (blue). Scale bars, 10 μm. Representative original data are shown. (**c**) Quantification of neutrophils in COVID–19 patients (*n* = 4) and control donors (*n* = 3) using an automatic digital slide scanner (Panoramic MIDI II, 3DHistech, Budapest, Hungary) is shown as cells/mm^2^. (**d**) Quantified fluorescence intensity sums of 4–HNE and MPO in COVID–19 patients (*n* = 4) and control donors (*n* = 3) were performed for each tissue section using Imaris software as intensity sum/µm^2^. The right panel shows the correlation between the intensity of 4–HNE and MPO in the lung tissues (*n* = 7). Values are means ± SEM. Single data are presented in scatter dot plots. ** *p* < 0.01; *** *p* < 0.001.

**Table 1 ijms-24-02646-t001:** Primer pairs used in this study.

Primer	Sequence (5′–3′)	Amplicon Size (bp)	T (°C)
*GAPDH* F	CAA CAG CCT CAA GAT CAT CAG CAA	103	60
*GAPDH* R	CAT GAG TCC TTC CAC GAT ACC		
*CRTAC1* F	ATC TTC TTC AAC AAC ATT GCC TAC	80	60
*CRTAC1* R	GGG TCT CCG TGC TCT CTA		
*COL1A1* F	GTT CGG AGG AGA GTC AGG AAG G	128	60
*COL1A1* R	CAG CAA CAC AGT TAC ACA AGG		

## Data Availability

The raw sequence data have been deposited in the Gene Expression Omnibus (GEO), accession number GSE208076. Requests for additional data presented in this study are available on request from the corresponding author.

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
