# Peer review of "Transcriptional Insights of Oxidative Stress and Extracellular Traps in Lung Tissues of Fatal COVID-19 Cases"

_ijms, 2023, doi:10.3390/ijms24032646_

Round 1

Reviewer 1 Report

The authors analyzed lung transcriptome from 7 patients with fatal COVID-19 compared with 3 samples from healthy donors. Strong changes were found for COL1A1 (up-regulation) and CRTAC1 (down-regulation) genes. Corresponding changes in oxidative stress and coronavirus infection, respectively, have been previously shown for these genes. Up-regulated genes were grouped into 10 top signaling pathways, which included, among others, genes involved in NET formation. A logical continuation of the study was to investigate MPO colocalization as an oxidative stress factor in neutrophils with oxidative stress markers on DNA (8-OHdG) and lipids (4-HNE). MPO levels were found to be more strongly associated with 4-HNE than MPO levels with 8-OHdG. This is a significant advantage of this study, since antibody staining does not unequivocally indicate pro-oxidative enzyme (MPO) activity, and in this case, there is direct evidence of oxidative stress marker formation and their colocalization with MPO. I have no global comments or question to authors of the manuscript.

In my opinion, it would be more convenient for readers if the figure or legends included the number of samples from patients and donors.

It would be fair to use the term reactive halogen species in addition to ROS, since it is the halogen atom in HOCl that is the oxidizing agent.

Author Response

The authors analyzed lung transcriptome from 7 patients with fatal COVID-19 compared with 3 samples from healthy donors. Strong changes were found for COL1A1 (up-regulation) and CRTAC1 (down-regulation) genes. Corresponding changes in oxidative stress and coronavirus infection, respectively, have been previously shown for these genes. Up-regulated genes were grouped into 10 top signaling pathways, which included, among others, genes involved in NET formation. A logical continuation of the study was to investigate MPO colocalization as an oxidative stress factor in neutrophils with oxidative stress markers on DNA (8-OHdG) and lipids (4-HNE). MPO levels were found to be more strongly associated with 4-HNE than MPO levels with 8-OHdG. This is a significant advantage of this study, since antibody staining does not unequivocally indicate pro-oxidative enzyme (MPO) activity, and in this case, there is direct evidence of oxidative stress marker formation and their colocalization with MPO. I have no global comments or question to authors of the manuscript.

In my opinion, it would be more convenient for readers if the figure or legends included the number of samples from patients and donors.

Answer: We thank the reviewer for his/her suggestion. The number of samples is added to the figure legends as requested.

It would be fair to use the term reactive halogen species in addition to ROS, since it is the halogen atom in HOCl that is the oxidizing agent.

Answer: We added a new reference and explanation for the role of reactive halogen species as requested by the reviewer (Reference: 60; page 12).

Reviewer 2 Report

The article is well constructed. Working methodology is well chosen and the results are accurate and clear presented and discussed !

Author Response

The article is well constructed. Working methodology is well chosen and the results are accurate and clear presented and discussed !

Answer: We thank the reviewer for the appreciation of our work.

Reviewer 3 Report

The manuscript by Hosseini, et al entitled “Transcriptional insights of oxidative stress and extracellular traps in lung tissues of fatal DOVID-19 cases” extends the recent studies examining the localization and constituents of neutrophil extracellular traps (NETs) in the pathophysiology of COVID-19 induced acute lung injury.  Many of the results from this study are expected, with a low level of novelty, however these results have not been previously reported and therefore add to the body of literature regarding the acute inflammatory response in the lung during COVID-19 infection. Results from this study show that there is an upregulation in the pathways regulating oxidative stress and viral mediated acute inflammation.  In addition, this study for the first time has revealed that NETS are associated with the neutrophil granule component myeloperoxidase, suggesting that the degree of neutrophil influx correlates with NET generation.  Finally, the manuscript shows that the lipid peroxidation product 4-hydroxynonenal (4-HNE) co-localizes with NETs in the lung and correlated with the number of neutrophils.  These are all interesting findings and further support that NET formation may play a significant role in the pathophysiology of acute lung inflammation after infection with COVID-19.

The main concern of the manuscript is the over representation of the results and potential therapeutic implications.  As only lung samples from deceased patients with COVID-19 were examined, it is unclear if similar findings are present in the lungs from patients less severely affected by COVID-19 infection or if the high level of NETs and lipid peroxidation products are only a marker of mortality.  In addition, as this was a one time point examination, it is not certain that NETs are present in the lung at earlier, or less severe, time points and more importantly if there is an excessive amounts of lipid peroxidation products present in the lung at these earlier time points, which is one potential treatment target proposed in the manuscript.  The statement that there was an excess neutrophil infiltration and NET formation in their patient is not supported by the Results as there is no comparison of neutrophil infiltration in the lungs of patients with less severe COVID-19 lung infection that survived in order to support these statements.  Finally, the comment that measurement of 4-HNE levels in the lung will be a useful biomarker for the severity of COVID-19 acute lung injury is impractical as lung tissue is not routinely obtained in patients with COVID-19 and particularly in those with severe disease.  

Author Response

The manuscript by Hosseini, et al entitled “Transcriptional insights of oxidative stress and extracellular traps in lung tissues of fatal DOVID-19 cases” extends the recent studies examining the localization and constituents of neutrophil extracellular traps (NETs) in the pathophysiology of COVID-19 induced acute lung injury.  Many of the results from this study are expected, with a low level of novelty, however these results have not been previously reported and therefore add to the body of literature regarding the acute inflammatory response in the lung during COVID-19 infection. Results from this study show that there is an upregulation in the pathways regulating oxidative stress and viral mediated acute inflammation.  In addition, this study for the first time has revealed that NETS are associated with the neutrophil granule component myeloperoxidase, suggesting that the degree of neutrophil influx correlates with NET generation.  Finally, the manuscript shows that the lipid peroxidation product 4-hydroxynonenal (4-HNE) co-localizes with NETs in the lung and correlated with the number of neutrophils.  These are all interesting findings and further support that NET formation may play a significant role in the pathophysiology of acute lung inflammation after infection with COVID-19.

The main concern of the manuscript is the over representation of the results and potential therapeutic implications.  As only lung samples from deceased patients with COVID-19 were examined, it is unclear if similar findings are present in the lungs from patients less severely affected by COVID-19 infection or if the high level of NETs and lipid peroxidation products are only a marker of mortality.  In addition, as this was a one time point examination, it is not certain that NETs are present in the lung at earlier, or less severe, time points and more importantly if there is an excessive amounts of lipid peroxidation products present in the lung at these earlier time points, which is one potential treatment target proposed in the manuscript.  The statement that there was an excess neutrophil infiltration and NET formation in their patient is not supported by the Results as there is no comparison of neutrophil infiltration in the lungs of patients with less severe COVID-19 lung infection that survived in order to support these statements. 

Answer: We accept the reviewer’s concern and revised the manuscript accordingly. We now specifically clarify that we compared the lungs of deceased COVID-19 patients with control donors. Also, as requested by the reviewer, we now mention the limitations of our study (page 13).

Finally, the comment that measurement of 4-HNE levels in the lung will be a useful biomarker for the severity of COVID-19 acute lung injury is impractical as lung tissue is not routinely obtained in patients with COVID-19 and particularly in those with severe disease. 

Answer: We thank the reviewer for his/her correction. We mention the use of 4-HNE plasma levels as a biomarker for the severity of infection (reference 44). We have now corrected this point (page 3).

Reviewer 4 Report

Authors have demonstrated an enriched NET formation pathway combined with LPO-derived reactive aldehydes and oxidized DNA damage in the lungs of severe COVID-19 patients.

The manuscript is well-written and interesting to read. The only concern I have is the use of RNA sequencing methods with archival formalin-fixed paraffin-embedded (FFPE) samples, which requires a reliable interpretation of the impact of pre-analytical variables on the data obtained, particularly the methods used to preserve samples and purify RNA. It would be useful to add a limitation section to the paper to elaborate on this issue and other limitations of this work.

Author Response

Authors have demonstrated an enriched NET formation pathway combined with LPO-derived reactive aldehydes and oxidized DNA damage in the lungs of severe COVID-19 patients.
The manuscript is well-written and interesting to read. The only concern I have is the use of RNA sequencing methods with archival formalin-fixed paraffin-embedded (FFPE) samples, which requires a reliable interpretation of the impact of pre-analytical variables on the data obtained, particularly the methods used to preserve samples and purify RNA. It would be useful to add a limitation section to the paper to elaborate on this issue and other limitations of this work.

Answer: Thank you for the suggestion. The limitation section is added in the Discussion of the manuscript (page 13).